# Micropulse cyclophotocoagulation compared to continuous wave cyclophotocoagulation for the management of refractory pediatric glaucoma

Bo Wang[1], Ryan T. Wallace[2], John A. Musser[2], Craig J. Chaya[2], Courtney L. Kraus[1]*

**1** Wilmer Eye Institute, Johns Hopkins Medicine, Baltimore, MD, United States of America, **2** John A. Moran Eye Center, Salt Lake City, UT, United States of America

* ckraus6@jhmi.edu

## Abstract

### Introduction

Micropulse cyclophotocoagulation (MPCPC) has been shown in adults to offer a favorable post-operative safety profile compared to continuous wave transscleral cyclophotocoagulation (CWCPC) in the management of glaucoma. The purpose of this study is to evaluate the long term efficacy, safety, and effectiveness of MPCPC in the management of pediatric glaucoma when compared to CWCPC.

### Methods

IRB approved retrospective chart review of patients with pediatric glaucoma that underwent MPCPC and CWCPC at 2 separate institutions. Success was defined as intraocular pressure (IOP) between 5 and 21mmHg on any number of topical glaucoma medication without requiring additional surgical intervention or oral IOP lowering medication.

### Results

Of the 48 patients in the study, 22 (26 eyes) underwent MPCPC and 26 (30 eyes) underwent CWCPC. At 1 year, 7 out of 26 eyes (26.9%) achieved success in the MPCPC group compared to 13 out of 30 eyes (43.3%) in the CWCPC group. Survival analysis unveiled a statistically significant difference in success between the two groups (p = 0.03).

### Conclusion

In pediatric glaucoma patients undergoing cyclophotocoagulation procedures, CWCPC outperformed MPCPC using default settings in terms of achieving long-term IOP control. Additional studies are required to evaluated augmented MPCPC settings in pediatric glaucoma patients.

**Data Availability Statement:** All relevant data are within the manuscript and its Supporting Information files.

**Funding:** The author(s) received no specific funding for this work.

**Competing interests:** BW, JM, RTW, and CK have no financial disclosures. CJC has received research and consulting honoraria from Abbvie and Ivantis, Inc. This does not alter our adherence to PLOS ONE policies on sharing data and materials.

**Abbreviations:** CWCPC, Continuous Wave Cyclophotocoagulation; CPC, Cyclophotocoagulation; IOP, Intraocular Pressure; MP, Micropulse; MPCPC, Micropulse Cyclophotocoagulation.

## Introduction

Cyclophotocoagulation (CPC) treats glaucoma by targeting and ablating the ciliary epithelium and stroma to decrease aqueous outflow, thereby lowering intraocular pressure (IOP) [1]. Continuous wave transscleral cyclophotocoagulation (CWCPC) is frequently used in the management of advanced glaucoma refractory to other treatments [2]. In recent years, an increasing number of publications reported that micropulse (MP) cyclophotocoagulation (CPC) offers a favorable post-operative safety profile compared to traditional continuous wave CWCPC in the management of refractory glaucoma in adults [3,4]. There are theoretical benefits of using MPCPC as it provides repetitive short pulses of energy followed by a period of rest. This reduces the high-intensity energy that is delivered to the ciliary body and is thought to reduce post-operative inflammation and complications [5]. Some reports even indicate more reliable sustained IOP reduction in 18 months of follow-up compared to CWCPC [4].

Despite multiple reports of sustained IOP reduction and the effectiveness of MPCPC in the management of adult glaucoma patients [6–12], there remains some controversy regarding the use of MPCPC, especially in early stage glaucoma patients [13]. In addition, only several studies [14–16] are currently available on the effectiveness of MPCPC in children, especially when compared to CWCPC. It is critical to assess the effectiveness of this technique as the application of MPCPC requires sedation in children, which negatively affects the benefit-to-risk ratio for the intervention. The purpose of this study is to investigate the IOP lowering effects and long-term outcomes of MPCPC compared to CWCPC at two different institutions.

## Methods

### Data collection

Both the John A. Moran Eye Center and Wilmer Eye Institute IRBs (IRB number—IRB 00263967) approved this retrospective study of pediatric (younger than 18 years of age) glaucoma patients that underwent MPCPC or CWCPC. The IRB at both institutions waived the need for signed consent for this retrospective review and the data was anonymized at each institution. Patient analysis spanned between September 2014 to January 2021 and all glaucoma subtypes were included: congenital glaucoma, anterior segment dysgenesis (Peters anomaly, aniridia), secondary glaucoma (trauma, phakomatoses, uveitis), glaucoma following cataract surgery, and juvenile open-angle glaucoma. All patients underwent general anesthesia for the procedure.

The decision to use MPCPC (MicroPulse P3) or CWCPC (G-Probe) with the Cyclo G6 Glaucoma Laser System (IRIDEX Corporation; Mountain View, CA, USA) was at the discretion of the attending surgeon. If patients had multiple MPCPC or CWCPC procedures, only the first procedure of its kind was included in the analysis. Power settings were at the discretion of the attending surgeon. MPCPC probe was firmly applied at the location of ciliary body and painted circumferentially at the limbus, with care taken to avoid the 3 and 9 o'clock meridian. When the limbus was challenging to identify, transillumination was used to assist in probe positioning. Average power for MPCPC was 1824±547 mW (range 1050–2000 mW) with a duration of 162±96 seconds (range 40–360 seconds), with a standard 31.3% duty cycle. The CWCPC probe was also firmly applied at the location of the ciliary body and individual applications were applied circumferentially at the limbus with care to avoid the 3 and 9 o'clock position. Average power for CWCPC was 1544±379 mW (range 700–2000 mW), 3.1±1.0 seconds (range 2–4 seconds) duration, and 17±5 shots (range 10–32 shots) at the discretion of the attending physician. The power and duration of the CWCPC was titrated down according to popping sounds. Visual field testing and optical coherence tomography (OCT) imaging were not recorded as end points of interest due to many younger patients' inability to tolerate

testing. Only patients with at least 1 year of follow-up or a surgical intervention due to uncontrolled IOP prior to the 1 year mark were included in the study.

## Outcomes

Success was defined as an IOP between 5 mmHg and 21 mmHg with or without glaucoma eye drops one year following surgery, but no oral glaucoma medications (i.e. Acetazolamide). For a patient to fail based on IOP alone, they needed 2 consecutive visits with recorded IOP greater than 21 mmHg. Having a subsequent glaucoma surgical intervention (i.e. tube shunt) was also considered a failure. Hypotony (IOP < 5mmHg) and phthisis bulbi were failure criteria as well. The primary outcome measure was the comparative success between the two procedures. Secondary analysis included the following variables: age, number of glaucoma medications, pre-operative IOP, glaucoma sub-type, laser power, and laser duration.

## Statistical analysis

Statistical analysis was performed using R (version 4.0.2, MAC). Demographic data and pre-operative data were analyzed using independent t-tests or Fischer exact tests. IOP differences between the pre-operative and post-operative pressure were analyzed using paired t-tests. Kaplan Meier survival plots were leveraged to analyze the survival between CWCPC and MPCPC and differences were compared using log-rank testing. A linear mixed effect model was used to assess the influence of pre-operative IOP, laser duration and power on whether a successful outcome was achieved following CWCPC or MPCPC.

## Results

A total of 26 eyes from 22 patients underwent MPCPC and a total of 30 eyes from 26 patients underwent CWCPC during the study period and fulfilled the study criteria. Baseline characteristics of the patient cohort can be found in Table 1. Patients that underwent CWCPC were

**Table 1. Baseline demographic and clinical data.**

|  | MPCPC (n = 26) | CWCPC (n = 30) | P-value |
|---|---|---|---|
| Age (years) | 7.0±4.4 | 8.8±5.6 | 0.167 |
| Preoperative IOP (mmHg) | 28.7±8.7 | 32.4±10.9 | 0.163 |
| Number of Medications (drops) | 2.77±1.07 | 2.76±1.30 | 0.994 |
| Oral IOP Medications | 2 (7.7%) | 5 (15.6%) | 0.389 |
| Number of Previous Glaucoma Surgeries | 1.5±0.8 | 1.3±1.4 | 0.856 |
| Glaucoma Diagnosis |  |  |  |
| Congenital Glaucoma | 11 (42%) | 8 (26.7%) |  |
| Anterior Segment Dysgenesis | 4 (15%) | 5 (16.7%) |  |
| Peters Anomaly | 2 | 3 |  |
| Aniridia | 0 | 2 |  |
| Axenfeld-Rieger Syndrome | 1 | 0 |  |
| Secondary Glaucoma | 8 (31%) | 7 (23.3%) |  |
| Uveitis | 0 | 2 |  |
| Phakomatoses | 5 | 1 |  |
| Trauma | 2 | 3 |  |
| Neovascular | 1 | 1 |  |
| Glaucoma Following Cataract Surgery | 3 (11%) | 9 (30.0%) |  |
| Juvenile Open-Angle Glaucoma | 0 (0%) | 1 (3.3%) |  |

P<0.05 is considered significant.

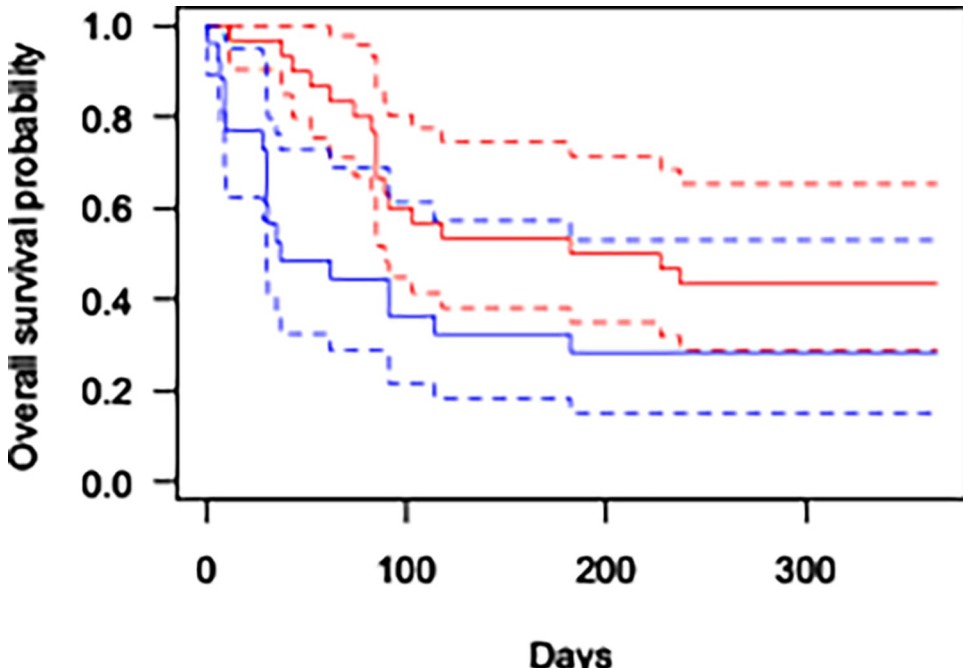

**Fig 1.** Kaplan-Meier survival analysis for success in both MPCPC (blue) and CWCPC (red). Dashed lines indicate the 95% confidence interval for each survival curve. P = 0.03.

typically older and more likely to be on oral IOP-lowering medication prior to undergoing cyclophotocoagulation.

The average pre-operative IOP was 28.7±8.7mmHg in the MPCPC group, with a post-operative 1-month average IOP of 23.3±9.7mmHg with a statistically significant difference before and after treatment (p = 0.011, 2 tailed). The average pre-operative IOP was 28.9±8.5mmHg in the CWCPC group, with a post-operative 1-month average IOP of 22.6±9.9mmHg with a statistically significant difference before and after treatment (p = 0.004, 2 tailed).

At 1-year post-op, 7 out of 26 eyes (26.9%) achieved success in the MPCPC group compared to 13 out of 30 eyes (43.3%) in the CWCPC group. Survival analysis demonstrated a statistically significant difference in success between the two groups (Fig 1, p = 0.03). There was no statistically significant change in the number of glaucoma drops patients were on before or after cyclophotocoagulation at 1-month post-op for either the CWCPC (2.2±1.5 drops) or the MPCPC group (2.3±1.1 drops).

There was no statistically significant effect of power, duration, or pre-operative IOP on the success of MPCPC (Fig 2). Similarly, for CWCPC, there was no significant effect of power, laser duration, or pre-operative IOP on surgical success (Fig 3). Average number of CWCPC shots for successful treatment was 15.9±5.0 shots and unsuccessful treatments was 18.3±4.9 shots (p = 0.21).

## Complications

Only one eye developed persistent hypotony after CWCPC for secondary glaucoma. No other incidence of hypotony or phthisis occurred during the study period. One eye from the MPCPC group was enucleated due to persistent pain and elevated IOP that was present prior to MPCPC, which did not improve after MPCPC. No eyes in the MPCPC group had long

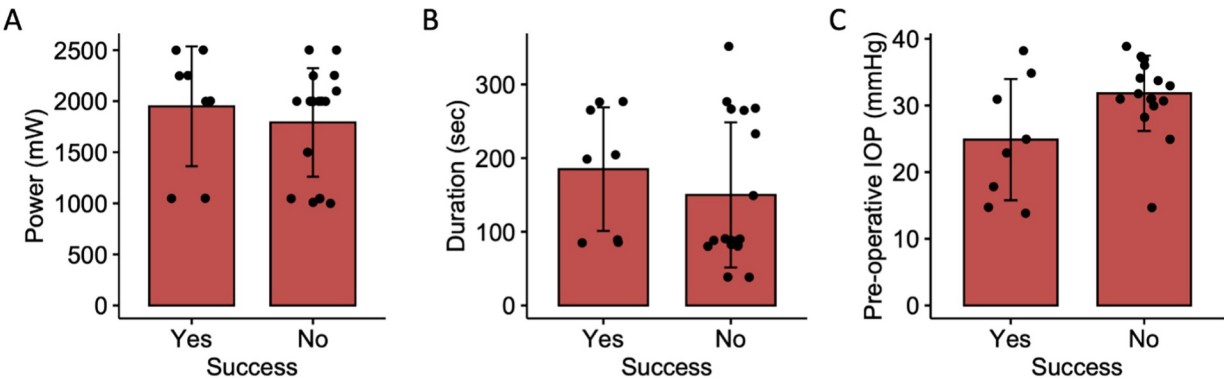

**Fig 2.** Bar plot illustrating the difference in A) power (mW), B) laser duration (seconds) and C) pre-operative IOP (mmHg) in whether IOP was successfully controlled with MPCPC.

term inflammation requiring steroid drops 2 months after the procedure. However, the CWCPC group had 2 eyes out of 30 requiring steroids for long term inflammation control.

## Discussion

MPCPC did provide initial IOP lowering after the surgery. However, MPCPC did not offer suitable long term sustained IOP lowering in our patient population. Only 7 out of 26 eyes (26.9%) experienced sufficient IOP reduction with MPCPC alone and did not require another surgical intervention or oral IOP-lowering medications. Those 7 successful patients included 3 patients (Fig 2C) who started off at normal (<21 mmHg) IOP, who received MPCPC due to concern that axial length continued to increase despite having an IOP < 21mmHg. It is likely that MPCPC is a safe choice in patients who need only a small degree of IOP lowering, which is not typically the case in the pediatric glaucoma population.

In contrast, greater than 40% of eyes undergoing CWCPC were able to achieve adequate IOP-lowering at 1-year post-operative follow-up. Given that MPCPC in a pediatric population requires the same degree of general sedation as conventional CWCPC, the potential benefits of less discomfort and ease of treatment reported in the adult population are unfortunately not as applicable to younger patient cohorts.

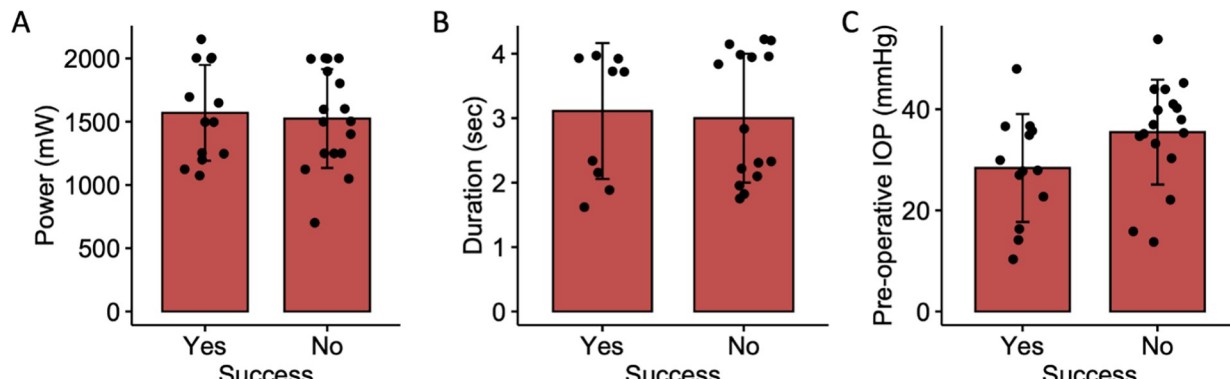

**Fig 3.** Bar plot illustrating the difference in A) power (mW), B) laser duration (seconds) and C) pre-operative IOP (mmHg) in whether IOP was successfully controlled with CWCPC.

There are several factors that may have played a role in the inferior outcome of MPCPC compared to CWCPC. These laser procedures were only performed in patients with refractory pediatric glaucoma, many of whom had several previous glaucoma surgeries. This cohort of patients could be less responsive to treatment compared to treatment-naïve children like those from the Lee et al. study for MPCPC [14], where 7 out of 9 patients had MPCPC performed as the first glaucoma surgical intervention of any kind.

There are significant differences in the settings used between the published pediatric studies on MPCPC. In general, the duty cycle of 31.3% is consistent across all studies due to the inherent 0.5 ms duration and 1.1 ms interval of the machine. However, there are significant treatment duration variations in the published literature ranging from approximately 60 seconds [15] to 160 seconds of laser active time [14]. In addition, the power used ranged from 1750-2000mW in various studies [14–16]. As this study incorporated two different institutions and multiple attending surgeons before the publication of augmented setting for MPCPC [17,18], there were significant variations in power and duration chosen, which are lower than the recently published guidelines. There was a trend towards higher power and the duration on the MPCPC being associated with greater success, but this was not statistically significant (Fig 2). Unfortunately, it is not possible to titrate the power of MPCPC during the procedure as is done for CWCPC by adjusting power based on audible sounds correlating to the destruction of the ciliary body. The lack of correlation between power and CWCPC is certainly consistent as laser power is typically titrated.

There are variations reported in the literature regarding the success of MPCPC in children. The study by Elhefney et al. allowed retreatment with MPCPC at the 2-month mark to achieve full success [15]. Ultimately, 66.7% of their study eyes were re-treated with MPCPC at the 2-month mark, which is consistent with our findings of failure (approximately two thirds of the patients) to adequately control IOP at the 2-month mark. Adult studies comparing MPCPC to CWCPC show excellent results with MPCPC [4,5], however, they also allowed retreatment and these findings may reflect differences in how adult and children respond to treatment. Abdelrahman et al reports an approximate 70% success rate with similar performance between MPCPC and CWCPC, with the CWCPC showing a worse side effect profile.[16] Future studies using augmented MPCPC [17] settings will likely show better performance with MPCPC than shown here.

While MPCPC did not have a strong sustained IOP-lowering effect, it did have low rates of residual post operative inflammation, as all patients were no longer requiring steroid drops by the 2-month mark post-operatively. There were 2 patients in the CWCPC group that had consistent low-grade inflammation after CPC even at the 6-month mark. This slightly elevated risk of post operative inflammation with CWCPC is consistent with prior adult [4] and pediatric studies [16] comparing the two techniques which suggest continuous laser energy is more likely to result in post-operative inflammation.

## Limitations

There are inherent limitations in this study given its retrospective nature and pediatric population. Further large, randomized trials would be necessary to demonstrate the long-term effectiveness of MPCPC in the pediatric population compared to CWCPC. Likewise, trials with standardized power settings, such as those recently advocated by Grippo et al. in their consensus guidelines, may result in optimized settings for pediatric patients [17]. However, these settings were not available during this study period. It was also difficult to truly assess intraocular inflammation post-procedure given the young ages of this cohort and difficulty with slit-lamp examination. Thus, persistent inflammation was generally confirmed via persistent injection

in the eye or pain. As many patients were unable to undergo OCT imaging or formal vision testing due to age, we did not report changes in visual acuity and long-term inflammation pathology such as macular edema secondary to cyclodestructive therapy.

## Conclusion

MPCPC did offer initial IOP lowering, which was unfortunately not sustained at the 1 year follow-up using the standard manufacturer settings. While CWCPC outperformed MPCPC in this study, further studies using the augmented MPCPC [17] settings are required to fully elucidate the role of MPCPC in the management of pediatric glaucoma.

## Supporting information

**S1 File.**
(PDF)

## Author Contributions

**Conceptualization:** Bo Wang, Craig J. Chaya, Courtney L. Kraus.

**Data curation:** Bo Wang, Ryan T. Wallace, John A. Musser, Craig J. Chaya, Courtney L. Kraus.

**Formal analysis:** Bo Wang, Courtney L. Kraus.

**Investigation:** Bo Wang, Courtney L. Kraus.

**Methodology:** Ryan T. Wallace, Courtney L. Kraus.

**Project administration:** Courtney L. Kraus.

**Resources:** Courtney L. Kraus.

**Supervision:** Courtney L. Kraus.

**Validation:** Bo Wang.

**Visualization:** Bo Wang, Courtney L. Kraus.

**Writing – original draft:** Bo Wang, Courtney L. Kraus.

**Writing – review & editing:** Bo Wang, Ryan T. Wallace, John A. Musser, Craig J. Chaya, Courtney L. Kraus.

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
