## [Decision Letter · Decision Letter 0]

1 Feb 2023

PONE-D-22-31494Micropulse Cyclophotocoagulation Compared to Continuous Wave Cyclophotocoagulation for the Management of Refractory Pediatric GlaucomaPLOS ONE

Dear Dr. Wang,

Thank you for submitting your manuscript to PLOS ONE. After careful consideration, we feel that it has merit but does not fully meet PLOS ONE’s publication criteria as it currently stands. Therefore, we invite you to submit a revised version of the manuscript that addresses the points raised during the review process.

We look forward to receiving your revised manuscript.

Kind regards,

Muhammad Qasim, Ph.D

Academic Editor

PLOS ONE

Journal Requirements:

In addition, as you are reporting a retrospective study of medical records or archived samples, please ensure that you have discussed whether all data were fully anonymized before you accessed them. If patients provided informed written consent to have data from their medical records used in research, please include this information.

“I have read the journal's policy and the authors of this manuscript have the following competing interests:

CJC has received research and consulting honoraria from Abbvie and Ivantis, Inc.”

Additional Editor Comments:

Reviewer 1:

The manscript "Micropulse Cyclophotocoagulation Compared to Continuous Wave Cyclophotocoagulation for the Management of Refractory Pediatric Glaucoma" is compiled to take in view to manage the pediatric glaucoma. over all, the manuscript conclude that MPCPC has short term effect on IOP lowering and CWCPC is more reliable than MPCPC.

I have only one question about this study. How would you justify your comparative study with a previous study as mentioned below

Micropulse versus continuous wave transscleral diode cyclophotocoagulation in refractory glaucoma: a randomized exploratory study in clinical & experimental ophthalmology 2015.

what do you think, which augmentation in MPCPC will become fruitful for the future studies.

Reviewer 2:

Manuscript is technically sound and data support conclusion. Data is available according to findings mentioned in manuscript but there is need to write about written consent of glaucoma patients about treatment in methods section.

Reviewers' comments:

Reviewer's Responses to Questions

**Comments to the Author**

1. Is the manuscript technically sound, and do the data support the conclusions?

Reviewer #1: Yes

Reviewer #2: Yes

2. Has the statistical analysis been performed appropriately and rigorously? 

Reviewer #1: Yes

Reviewer #2: Yes

3. Have the authors made all data underlying the findings in their manuscript fully available?

Reviewer #1: Yes

Reviewer #2: Yes

4. Is the manuscript presented in an intelligible fashion and written in standard English?

Reviewer #1: Yes

Reviewer #2: Yes

5. Review Comments to the Author

Reviewer #1: The manscript "Micropulse Cyclophotocoagulation Compared to Continuous Wave Cyclophotocoagulation for the Management of Refractory Pediatric Glaucoma" is compiled to take in view to manage the pediatric glaucoma. over all, the manuscript conclude that MPCPC has short term effect on IOP lowering and CWCPC is more reliable than MPCPC.

I have only one question about this study. How would you justify your comparative study with a previous study as mentioned below

Micropulse versus continuous wave transscleral diode cyclophotocoagulation in refractory glaucoma: a randomized exploratory study in clinical & experimental ophthalmology 2015.

what do you think, which augmentation in MPCPC will become fruitful for the future studies.

Reviewer #2: Manuscript is technically sound and data support conclusion. Data is available according to findings mentioned in manuscript but there is need to write about written consent of glaucoma patients about treatment in methods section.

6. PLOS authors have the option to publish the peer review history of their article (what does this mean?). If published, this will include your full peer review and any attached files.

Reviewer #1: No

Reviewer #2: No

---

## [Author Response · Author response to Decision Letter 0]

20 Mar 2023

We thank both reviewers for their excellent comments and feedback. Please find below a point-by-point response of their comments below. During data upload, it was noted that 2 patients in the TSCPC group who were excluded during the data analysis (insufficient follow up) were accidentally counted in Table 1. The table and abstract were updated to reflect that exclusion and there is no change to the results or conclusion of the paper. 

Reviewer 1:

The manscript "Micropulse Cyclophotocoagulation Compared to Continuous Wave Cyclophotocoagulation for the Management of Refractory Pediatric Glaucoma" is compiled to take in view to manage the pediatric glaucoma. over all, the manuscript conclude that MPCPC has short term effect on IOP lowering and CWCPC is more reliable than MPCPC.

I have only one question about this study. How would you justify your comparative study with a previous study as mentioned below

Micropulse versus continuous wave transscleral diode cyclophotocoagulation in refractory glaucoma: a randomized exploratory study in clinical & experimental ophthalmology 2015.

what do you think, which augmentation in MPCPC will become fruitful for the future studies.

We completely agree that augmentation in MPCPC could potentially improve the capability of MPCPC to lower IOP. Future work is required to evaluate these enhanced settings. “Likewise, trials with standardized power settings, such as those recently advocated by Grippo et al. in their consensus guidelines, may result in optimized settings for pediatric patients.17”

The published study shows extraordinary results of MPCPC compared to continuous wave laser, with power settings like ours. There are several differences that can account for this, the authors allowed retreatment (half the patients received retreatment) and the study was performed in adults, who might respond differently than children to micropulse. We added: “Adult studies comparing MPCPC to CWCPC show excellent results with MPCPC, however, they also allowed retreatment and these findings may reflect differences in how adult and children respond to treatment.”

Reviewer 2:

Manuscript is technically sound and data support conclusion. Data is available according to findings mentioned in manuscript but there is need to write about written consent of glaucoma patients about treatment in methods section.

We added the following to the methods section: “The IRB at both institutions waived the need for signed consent for this retrospective review and the data was anonymized at each institution.”

---

## [Decision Letter · Decision Letter 1]

9 Aug 2023

PONE-D-22-31494R1Micropulse Cyclophotocoagulation Compared to Continuous Wave Cyclophotocoagulation for the Management of Refractory Pediatric GlaucomaPLOS ONE

Dear Dr. Wang,

Thank you for submitting your manuscript to PLOS ONE. After careful consideration, we feel that it has merit but does not fully meet PLOS ONE’s publication criteria as it currently stands. Therefore, we invite you to submit a revised version of the manuscript that addresses the points raised during the review process.

ACADEMIC EDITOR:The authors have responded to the previous review comments adequately. There remain a few additional minor concerns. Please review the current reviewers' comments.==============================

We look forward to receiving your revised manuscript.

Kind regards,

Nader Hussien Lotfy Bayoumi, M.D., FRCS (Glasgow)

Academic Editor

PLOS ONE

Journal Requirements:

Reviewers' comments:

Reviewer's Responses to Questions

**Comments to the Author**

1. If the authors have adequately addressed your comments raised in a previous round of review and you feel that this manuscript is now acceptable for publication, you may indicate that here to bypass the “Comments to the Author” section, enter your conflict of interest statement in the “Confidential to Editor” section, and submit your "Accept" recommendation.

Reviewer #3: (No Response)

Reviewer #4: All comments have been addressed

2. Is the manuscript technically sound, and do the data support the conclusions?

Reviewer #3: Partly

Reviewer #4: Yes

3. Has the statistical analysis been performed appropriately and rigorously? 

Reviewer #3: I Don't Know

Reviewer #4: Yes

4. Have the authors made all data underlying the findings in their manuscript fully available?

Reviewer #3: No

Reviewer #4: Yes

5. Is the manuscript presented in an intelligible fashion and written in standard English?

Reviewer #3: Yes

Reviewer #4: Yes

6. Review Comments to the Author

Reviewer #3: The present study compared the one-year results of MPCPC in the management of pediatric glaucoma when compared to CWCPC and concluded that CWCPC outperformed MPCPC.

Generally, the study results enrich the literature in this controversy. As both techniques are compared, it must be clear that there are many variables that affect the outcome, including: the recruited cases, previous surgeries, surgical technique and settings, surgeon’s experience, disease stage, gender, number of treatments, racial differences, etc.

I have the following comments:

Lines 26-27: are the one-year results considered long term?

Line 45: to “evaluate” instead of “evaluated “

Line 74: the statement may give the impression of comparing the results in two different institutions.

Any exclusion criteria? are the cases classified as congenital glaucoma recurrent or de novo cases?

Lines 87-95: great variations in the treatment settings without any obvious guidelines. e.g., a wide range of MPCPC treatment durations (40-360 seconds) that would affect the outcome.

No mention of the anesthesia used.

No mention of the detailed fine surgical details in both groups.

Line 100: would the authors consider re-treatment a failure? This is not clear in the manuscript.

Lines 122- 124: Patients that underwent CWCPC were typically older and more likely to be on oral IOP-lowering medication prior to undergoing cyclophotocoagulation. This contradicts the statement at lines 87-89.

Reviewer #4: Comments to authors:

The manuscript deals with an important topic and covers an area of research not

sufficiently covered in previous research. The authors conducted a retrospective study

to compare "Micropulse Cyclophotocoagulation to Continuous Wave

Cyclophotocoagulation for the Management of Refractory Pediatric Glaucoma" The

manuscript concludes that in pediatric glaucoma patients undergoing

cyclophotocoagulation procedures, CWCPC outperformed MPCPC using default

settings in terms of achieving IOP control for at least one year.

Overall; the paper is well-written and organized. Here below; some comments:

Introduction

This section contains enough general and specific background information but we

need more description of the gap in our knowledge that the study was designed to fill.

Methodology

1)-The authors have to mention some details about the instruments used in the clinical

examination of the patients, especially the IOP measuring device and whether IOP

measurements were done under general anaesthesia or not, and what type of

anaesthesia used

2)-The authors have to mention some details about how the Cyclophotocoagulation

probe was applied in both groups e.g

MP-CPC Group:

• The probe was applied with firm pressure, perpendicular to the sclera, with the

probe edge ……etc

• Avoiding the 3 and 9 o’clock meridians?

CW-CPC Group

• Transillumination with a light source in a darkened theater to accurately locate

the position of the ciliary body was used or not?

• The power and duration of the laser application were increased until a popping

sound was heard. Is this method followed or not?

• The number of popping shots?

• Avoiding 3 and 9 o’clock positions

Discussion

This section was more or less well-written with a clear comparison to the previously

published results, however, it was missing a detailed comparison with the study of

Abdelrahman & El Sayed (16) which is the nearest to the subject of the present study

7. PLOS authors have the option to publish the peer review history of their article (what does this mean?). If published, this will include your full peer review and any attached files.

Reviewer #3: No

Reviewer #4: No

---

## [Author Response · Author response to Decision Letter 1]

22 Aug 2023

We would like to thank both reviewers for their insightful comments regarding the manuscript. We feel that the manuscript has been substantially improved as a result, and hope that it will now be acceptable for publication in PLOS ONE. Please find below a point-by-point response to the feedback from the Reviewers. 

Comments to authors:

The manuscript deals with an important topic and covers an area of research not sufficiently covered in previous research. The authors conducted a retrospective study to compare "Micropulse Cyclophotocoagulation to Continuous Wave Cyclophotocoagulation for the Management of Refractory Pediatric Glaucoma" The manuscript concludes that in pediatric glaucoma patients undergoing cyclophotocoagulation procedures, CWCPC outperformed MPCPC using default settings in terms of achieving IOP control for at least one year.

Overall; the paper is well-written and organized. Here below; some comments:

Introduction

This section contains enough general and specific background information but we need more description of the gap in our knowledge that the study was designed to fill.

Methodology

1)-The authors have to mention some details about the instruments used in the clinical examination of the patients, especially the IOP measuring device and whether IOP measurements were done under general anaesthesia or not, and what type of anaesthesia used.

As this was a retrospective study across two different institutions. IOP measurement were done based on the preference of the institution and not standardized. However, the trend of improvement or lack thereof should not be significantly impacted on this.

We added: “All patients underwent general anesthesia for the procedure.”

2)-The authors have to mention some details about how the Cyclophotocoagulation probe was applied in both groups e.g

MP-CPC Group:

• The probe was applied with firm pressure, perpendicular to the sclera, with the probe edge ......etc

• Avoiding the 3 and 9 o’clock meridians? 

We added: “MPCPC probe was firmly applied at the location of ciliary body and painted circumferentially at the limbus, with care taken to avoid the 3 and 9 o’clock meridian. When the limbus was challenging to identify, transillumination was used to assist in probe positioning.” 

CW-CPC Group

• Transillumination with a light source in a darkened theater to accurately locate the position of the ciliary body was used or not?

• The power and duration of the laser application were increased until a popping sound was heard. Is this method followed or not?

• The number of popping shots?

• Avoiding 3 and 9 o’clock positions

Unfortunately, due to this being a retrospective study across two institutions, there was not a standardized method performed to located the ciliary body. The number of popping shots vs non-popping shots were not recorded in the charts. 

We added: “The CWCPC probe was also firmly applied at the location of the ciliary body and individual applications were applied circumferentially at the limbus with care to avoid the 3 and 9 o’clock position…The power and duration of the CWCPC was titrated down according to popping sounds.”

Discussion

This section was more or less well-written with a clear comparison to the previously published results, however, it was missing a detailed comparison with the study of Abdelrahman & El Sayed (16) which is the nearest to the subject of the present study.

We added: “There are variations reported in the literature regarding the success of MPCPC in children… Abdelrahman et al reports an approximate 70% success rate with similar performance between MPCPC and CWCPC, with the CWCPC showing a worse side effect profile.”

---

## [Editor Report · Decision Letter 2]

25 Aug 2023

Micropulse Cyclophotocoagulation Compared to Continuous Wave Cyclophotocoagulation for the Management of Refractory Pediatric Glaucoma

PONE-D-22-31494R2

Dear Dr. Wang,

We’re pleased to inform you that your manuscript has been judged scientifically suitable for publication and will be formally accepted for publication once it meets all outstanding technical requirements.

Kind regards,

Nader Hussien Lotfy Bayoumi, M.D., FRCS (Glasgow)

Academic Editor

PLOS ONE

Additional Editor Comments (optional):

Thank you for fulfilling the reviewers' comments.
---

## [Editor Report · Acceptance letter]

20 Dec 2023

PONE-D-22-31494R2 

PLOS ONE

Dear Dr. Wang, 

I'm pleased to inform you that your manuscript has been deemed suitable for publication in PLOS ONE. Congratulations! Your manuscript is now being handed over to our production team.

Kind regards, 

on behalf of

Professor Nader Hussien Lotfy Bayoumi 

Academic Editor

PLOS ONE